# A study on the impact of enterprise digital transformation on informed trading

**Hualing Wang** [ORCID] *

School of Economics and Management, Quanzhou University of Information Engineering, Quanzhou, Fujian, China

* wanghualing@qzuie.edu.cn

## Abstract

Informed trading, driven by information asymmetry and market imperfections, varies in presence across markets. This form of trading not only distorts market transaction prices and hinders resource allocation but also initiates adverse selection transactions, increasing liquidity risks and potentially precipitating market crashes, thereby impeding the market's healthy development. Utilizing information asymmetry theory and principal-agent theory, this paper analyzes data from A-share listed companies from 2011 to 2022. Employing a fixed-effect model, it empirically examines the influence of enterprise digital transformation on the likelihood of informed trading. The findings demonstrate that enterprise digital transformation markedly reduces the likelihood of informed trading. Further analysis of heterogeneity indicates that, compared to state-owned, non-high-tech enterprises and enterprises in the western region, the inhibitory effect on informed trading is more pronounced in non-state-owned, high-tech enterprises and enterprises in the eastern and central regions. Additionally, the chain mediation effect underscores that digital transformation weakens information asymmetry and strengthens internal controls, thereby reducing informed trading. Finally, employing a dynamic panel threshold model we find that digital transformation can only significantly inhibit the informed transactions when enterprises have reached a certain level of technological and asset accumulation.

**Data Availability Statement:** All relevant data are within the manuscript and its Supporting Information files.

**Funding:** This research is supported by Fujian Social Science Fund Project (FJ2024B035), and the funders had no role in study design, data collection

## 1. Introduction

In micro markets, there are varying degrees of informed trading due to manifestations of market failure and information asymmetry. Informed traders gain a significant information advantage, allowing them to secure substantial excess returns in market transactions—returns that constitute losses for uninformed traders [1–3]. This phenomenon of securing excess returns through concealed information exists not only in mature markets such as the U.S. [4], the UK [5], and Germany [6], but is even more prevalent in less mature emerging markets. For example, in China's A-share market, company executives demonstrate a strong capacity to time transactions in their company's shares, earning significant gains. Informed individuals not only manipulate and distort market prices, thereby undermining the market pricing mechanism and reducing market efficiency [7], but they also trigger adverse selection, increasing

and analysis, decision to publish, or preparation of the manuscript.

**Competing interests:** The authors have declared that no competing interests exist.

market liquidity risk and exacerbating market failure, which may ultimately lead to market crashes [8]. The rampant increase in informed trading and insider trading poses a severe threat to investors' rights to equal information and property, and to the smooth functioning of the market. Therefore, in the era of rapid digital technological advancement, studying the impact of corporate digital transformation on informed trading behavior is of significant theoretical and practical value.

Information as an economic good can mitigate or eliminate the problems of moral hazard and adverse selection within the enterprise. Open information within enterprises not only reduces financing costs but also prevents management (informed traders) from concealing negative corporate news [9, 10]. However, identifying informed trading empirically is challenging because the information sources of informed traders are unobservable, and they often disguise their trading activities by concealing these sources [11]. The prevalence of hidden trading behaviors is partly due to internal control deficiencies within companies, where existing managerial agency issues provide opportunities for such concealment [12]. Managers can easily circumvent internal controls over information and communication, exploiting supervision loopholes to conceal their trading activities, thereby facilitating the concealment of adverse news [13]. The American Loyalty and Guaranty Corporation's findings also indicate that most corporate bankruptcies stem from ineffective internal controls [14]. In firms, deficiencies in internal control may manifest as irrational organizational structures and ambiguous distribution of authority and responsibility, allowing some individuals to bypass organizational rules. This can lead to opaque corporate governance, inefficient information communication, and facilitated transaction concealment [15]. In companies with less effective internal controls, managers can secure greater operational liberties, making it easier for informed traders (managers or related subjects) to hide their activities [16]. Another reason for the tendency of informed traders to conceal their activities is the information advantage they hold. Information varies in its exclusivity; it can be public market information, which is non-exclusive, or it can derive from specialized private channels that offer significant exclusivity in terms of access, costs, and benefits [17]. Uninformed traders without access to private information channels may consider purchasing information, thus becoming informed traders [18]. However, in emerging markets, higher information search costs discourage market participants from extracting profitable, firm-specific information [19]. These constraints on information transmission channels and associated costs create disparities in information access among market players. Consequently, managers with the prerogative for abnormal operations are more likely to access differential information and engage in informed trading.

The process of information from generation to full assimilation by the market encompasses the spread of relatively objective information across markets and the precise comprehension of relatively subjective information content by individuals [20], i.e., the degree of information disclosure and the accuracy of information expectations. Increasing the availability of public information in the market enhances price discovery, diminishes the importance of private information, and directs the rational allocation of resources, thereby reducing the likelihood of informed trading [18]. In addition, augmented public information decreases the transparency in capital cost for firms and lowers the expense of acquiring private information for general traders [21]. As a pivotal mechanism for public information, stringent disclosure requirements can lessen discrepancies among market participants, improve the regulatory efficacy of authorities, curb the opportunistic actions of financial professionals (informed traders), and bolster the relative fairness of information in market transactions. Thus, information disclosure is pivotal to the interests of numerous investors and reflects market fairness and efficiency [22]. The accuracy of information predictions primarily stems from the expertise of the predictor and their access to private information. Due to the substitution effect between public and private

information [23], an information disclosure system or technology can facilitate the conversion of private information into public information and enhance the level of disclosure [24]. Although information disclosure does not necessarily improve the situation for all investors in a firm [25, 26], it can reduce the proportion of private information, amplify the impact of public information on the market price discovery mechanism [27], and encourage market participants to focus more on public information and enhance their analytical capabilities.

From the perspective of external influence, technology empowerment and data-driven practices within the digital economy enhance the quality of information disclosure, which is influenced by the channels through which it is disclosed. Under the support of digital technology, the output of enterprise information becomes more standardized, thereby likely improving the quality of market information disclosure. Additionally, general market participants (non-informed traders) benefit from an increased supply of standardized information, which reduces the costs of information acquisition and identification, and improves the fairness of transaction information [28]. Moreover, digital technology can also suppress the supply of noise information and enhance the quality of public information, especially representational information on the website and social media [29]. The prevalence of noise information impairs the ability of market participants to filter information effectively, while the digital transformation of enterprises significantly enhances data processing and application [30], weakens information asymmetry, ensures the authenticity of information, and reduces the impact of noise.

From an internal impact analysis, digital transformation directly influences a company's operational efficiency, performance, information disclosure, and corporate governance [31]. Digital technology enables enterprises to access more information, which can be used to analyze and predict managerial behaviors, thereby facilitating the regulation of these behaviors and reducing agency and monitoring costs [32]. The adoption of digital technology also encourages shareholder participation and enhances the transparency of internal controls [33]. Improvements in the efficiency of corporate internal controls strengthen the reliability of corporate information, reduce the information advantage held by insiders (informed traders) [34], and inhibit informed trading.

Therefore, based on the above theoretical deduction, this study posits that information asymmetry in the market inevitably leads to the existence of informed trading, which severely undermines the market price guidance mechanism and leads to market failure. However, the covert nature of informed trading complicates its supervision and management by the market. Digital transformation can enable enterprises to leverage digital technology to expand information access channels, enhance information transparency, mitigate the principal-agent dilemma caused by information asymmetry, and curb informed trading behavior. This research aims to contribute to and innovate within the literature by exploring, from the perspective of micro-market structure theory, how enterprises can enhance the efficiency and level of resource management, improve both internal and external information transmission channels, and reduce the likelihood of informed trading through digital transformation. This approach not only enriches and deepens the theory related to the inhibition of informed trading by digital transformation but also introduces an intermediary index of enterprise internal control and constructs a chain intermediary effect model. This method enriches the research methodology of informed trading and, based on the principal-agent theory, delves into how enterprise digital transformation indirectly affects the likelihood of informed trading by influencing information asymmetry levels and optimizing internal control. Lastly, considering the variability in the accumulation of digital technologies and enterprise sizes, and acknowledging the presence of threshold effects in these areas, this study tests these differences using a threshold effect model. The findings of this study are not only of significant theoretical relevance but also offer practical guidance for enterprise digital transformation.

## 2. Literature review and hypothesis formulation

In the domain of market trading, the concepts of informed trading and information asymmetry are pivotal in shaping trading behaviors and market dynamics. According to the information asymmetry perspective of market microstructure theory, market traders are differentiated into informed traders, who possess private information, and uninformed traders, who lack such information [35]. During trading activities, uninformed traders require higher risk premiums due to their informational disadvantage, leading to adverse selection in the market [36]. Conversely, informed traders, equipped with private information, achieve higher returns through additional information [18]. The heterogeneous nature of traders' information resources allows informed traders to drive the trading price closer to the fundamental value using their information advantage but also to mislead non-informed traders and profit from them through the dissemination of noisy information, which significantly disrupts market order [37].

As the cornerstone of modern economic theory, the efficient market hypothesis (EMH) posits that for markets to be considered efficient, investors must be fully rational, possess uniformly accessible, non-discriminatory information, and be fully informed [38]. However, the realities of information acquisition costs and price distortions indicate that the EMH often fails to hold [18]. Markets exhibit basic characteristics such as differing capabilities in information processing, sequential information access, and varied access to insider information [39]. Consequently, informed traders, leveraging their informational advantage, often secure excess returns by acting prior to public information announcements—returns that essentially compensate for their information acquisition costs [18]. Maffett [40] explored the link between financial reporting opacity and informed trading among international institutional investors, discovering that firms within more opaque informational environments are prone to increased incidences of private informed trading. Systematic and significant information asymmetry among investors can elevate transaction costs, dampen market efficiency, and reduce liquidity, particularly in emerging capital markets where investor protection is lax and the cost of accessing private information is high. This environment fosters informed trading and heightens the risk of noise trading [19]. Market noise may stem from investors' cognitive limitations or deficiencies in information processing technology. However, digital transformation can weaken information asymmetry and mitigate the negative impacts of noise trading [41].

Digital transformation represents a process wherein companies utilize digital technology to enhance value creation and adapt to changes in the external environment. With advancements in information technologies such as big data, digital transformation has facilitated more accurate forecasts and expedited information feedback [42]. From an internal information management perspective, digital transformation standardizes information processing, ensuring information is immediate, transparent, and verifiable. This standardization makes it challenging for management to manipulate information and enhances internal control mechanisms [43, 44], transitioning private information into the public domain, thus reducing market players' information collection costs and improving information availability [45]. From the standpoint of signal theory, while it is challenging for enterprises to balance the effective dissemination of information with the protection of commercial secrets [46], digital transformation standardizes the channels through which information is released. This standardization facilitates the emission of more effective signals, increases the share of public information in the market, allows market players to better filter out noise, and enhances the credibility and transparency of public information. Ultimately, this empowerment of internal management leads to a reduction in informed trading [47]. Based on the foregoing research, the following hypotheses is proposed:

**Hypothesis 1:** Enterprise digital transformation significantly reduces the likelihood of informed trading in the market.

According to the theory of information asymmetry, external investors cannot fully comprehend the business dynamics of an enterprise, and management may exploit this information disparity to diminish the quality of the enterprise's public disclosures [10]. Information asymmetry combined with agency problems often enables managers to evade regulatory constraints, leading to internal control deficiencies [48], which in turn increase the likelihood of informed management transactions. Digital transformation within enterprises can mitigate these issues.

Digital transformation aids in reducing information asymmetry, curbing internal control shortfalls, and decreasing the likelihood of informed trading. Ashbaugh-Skaife, Collins [15] identified the lack of access to sufficient and effective information as a major factor contributing to internal control deficiencies. Digital transformation enhances information disclosure and weakens information asymmetry by improving information channels, augmenting the information available for corporate decision-making, and enhancing decision-making efficiency. Primarily, the digitalization of enterprises optimizes information acquisition channels and alleviates information bias. The application of digital technologies significantly enhances an enterprise's capacity to process unstructured and non-standardized data, transforming it into structured and standardized information, thereby optimizing information acquisition channels [49]. Furthermore, digital technologies like blockchain enrich the decision-making information necessary for controlling enterprise activities and reduce the conditions conducive to informed transactions [50]. Moreover, enterprise digital transformation facilitates a "digital" representation of business activities, supplying more information for internal control decision-making. It enables enterprises to utilize efficient budget data for effective internal control [51]. Additionally, digitalization of internal management processes allows for the clear display of complex management activities, assisting control entities in precisely evaluating the internal information necessary for decision-making on control activities and enhancing the efficiency of internal controls. Lastly, digital technology also supports enterprises in evaluating the effectiveness of decision-making methods and improving decision-making efficiency. By enhancing the adequacy and effectiveness of decision-making information required for control activities, digital transformation enables enterprises to adopt diverse decision-making strategies, better align internal and external information, and fully leverage the value of information. It also aids control entities in utilizing enterprise risk assessment systems to provide timely feedback to information users, thereby enhancing the effectiveness of risk response strategies.

On the other hand, enterprises can mitigate agency conflicts between shareholders and managers through digital transformation, reduce internal control deficiencies, and decrease the likelihood of informed trading. The central role of managers within an enterprise's internal control system can precipitate agency issues and engender weaknesses in internal control. Digital transformation enhances communication and intensifies oversight to curtail internal control deficiencies. Initially, digital transformation boosts communication efficiency. The use of digital technologies enables shareholder participation in the enterprise's decision-making processes, thereby improving the effectiveness of communication and preventing managers from exploiting their authority to conduct informed transactions that harm shareholder interests. Furthermore, digital transformation encourages enterprises to continuously refine their organizational structure and elevate management efficiency, thereby circumventing internal control norms [52]. It promotes a flatter organizational structure, enabling better cooperation and collaboration across functional departments. The use of digital technology for data sharing

clarifies managerial authority and improves the precision and effectiveness of standard implementation and control activities, thus preventing the distortion of these activities by powerful executives [53–55]. By reshaping the organizational structure, enterprises optimize internal control processes and further mitigate control weaknesses through digital transformation [56]. Additionally, digital transformation diminishes managerial power by amplifying the influence of grassroots levels and enhancing external oversight, thus limiting managers' discretionary control over business activities and curtailing informed trading by managers [57].

**Hypothesis 2:** Enterprise digital transformation enhances the information transparency of the enterprise, which in turn strengthens the enterprise's internal control capabilities and reduces the probability of informed trading.

According to Porter [58], "For a nation's firms to compete effectively in the international marketplace, they must continually innovate to enhance their competitive advantage." In today's uncertain and complex world, innovation is increasingly crucial for organizations [59]. The innovation capability of an organization is derived from the accumulation of knowledge and technology [60–62]. The law of accelerated returns posits that technological innovation is based on previous achievements, and its exponential impact on the economy depends on continuous technological accumulation. Past accumulations of technological innovations significantly increase the efficiency of digital technological transformations. Hence, technological accumulation enhances the efficiency of digital transformation by upgrading digital innovation skills and personnel technological literacy, providing optimal conditions for information disclosure, and reducing the likelihood of informed trading. Once enterprises achieve a certain level of technological accumulation, digital transformation enables the replication and dissemination of knowledge and information at near-zero marginal cost [63], diminishes the time and space barriers to information transfer, increases the frequency of information dissemination [64], and decreases the probability of informed trading. Moreover, as the scale of enterprise technology accumulation grows, enterprise technicians can further analyze and apply existing technologies by reviewing the technology accumulation process, mastering technological relationships in-depth, and enhancing technological literacy [65]. The improvement in technicians' literacy promotes the digital transformation of enterprises, improves the level of information disclosure, and curtails informed trading. Consequently, the following hypothesis is formulated:

**Hypothesis 3a:** As technology accumulation increases, the inhibitory effect of enterprise digital transformation on informed trading intensifies.

High-quality information disclosure enables investors and the public to better understand the operations of an enterprise, and accurate information disclosure plays a crucial role in the capital market's resource allocation. Dierkes and Coppock [66] discovered through empirical research that company size and the level of disclosure are significantly positively correlated. Berger and Udell [67] also noted that SMEs inherently have a weaker capacity for producing digital information, but this capacity increases with firm size. Therefore, as firm size expands, the internal institutions of firms improve, the conditions for supplying standardized digital information become more adequate, and the more standardized the corporate information disclosure, the more pronounced the inhibition of informed trading by digital transformation becomes. Therefore, the following hypothesis is proposed:

**Hypothesis 3b:** As enterprise scale increases, the inhibitory effect of enterprise digital transformation on informed trading is enhanced.

The framework diagram for this paper is shown below (Fig 1):

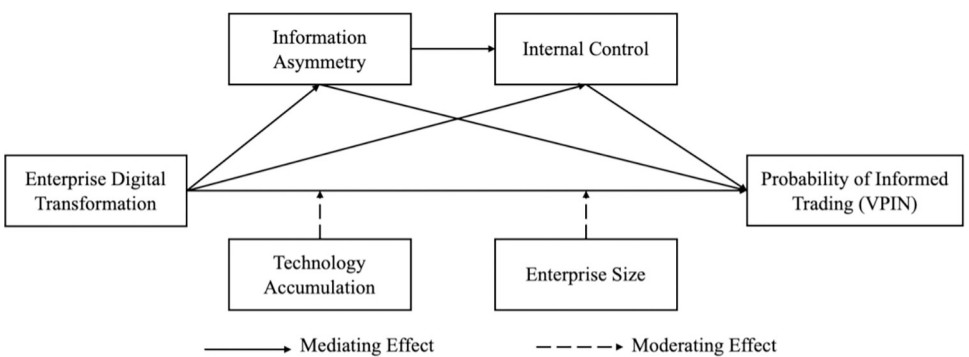

**Fig 1. Framework diagram.**

## 3. Research design

### 3.1. Sample selection

This study examines all A-share listed companies from 2011 to 2022 as the initial research subjects. The data treatment involves several steps: firstly, excluding non-ordinary trading companies (including ST, ST*, and PT); secondly, removing companies with a gearing ratio less than 0 or greater than 1; thirdly, omitting companies with a significant number of missing data points in relevant variables; and fourthly, excluding companies within the financial sector. The primary data for variables such as the enterprise digital transformation index and the probability of informed trading are sourced from the CSMAR database. After processing and eliminating companies with extensive missing data, we secured a total of 20064 annual observations.

### 3.2. Variable measurement

**3.2.1. Explained variables.** Probability of Informed Transaction (*VPIN*) is calculated based on the ratio of the informed transaction order arrival rate to the total order arrival rate, employing the BVC algorithm for estimation as per [8]. Let the volume of buy and sell transactions be respectively defined:

$$\begin{cases} V_\zeta^B = \sum_{i=t(\zeta-1)+1}^{t(\zeta)} V_i \times Z\left(\dfrac{P_i - P_{i-1}}{\sigma_{\triangle P}}\right) \\ V_\zeta^S = \sum_{i=t(\zeta-1)+1}^{t(\zeta)} V_i \times \left[1 - Z\left(\dfrac{P_i - P_{i-1}}{\sigma_{\triangle P}}\right)\right] \end{cases} \tag{1}$$

where t ($\zeta$) is the end of the trading basket, i denotes the shortest time interval (e.g., 1min), $V_i$ represents the trading volume at time i, $P_i$ indicates the price at time i, $\sigma_{\triangle P}$ is the standard deviation of the volume of all price changes within each basket, and Z represents the cumulative distribution function of the standard normal distribution. Based on this, VPIN can be expressed as:

$$VPIN = \frac{\sum_{\zeta=1}^{n} |V_\zeta^B - V_\zeta^S|}{nV} \times 100 \tag{2}$$

N denotes the number of trading baskets within the interval, and V indicates the volume of each basket. In this study, we use n = 8 as the number of daily trading baskets, and the minimum time interval i = 1. The static estimation method treats daily trading as the interval, using the arithmetic mean of daily VPIN values throughout the year as the annual VPIN.

**3.2.2. Explanatory variables.** Enterprise digital transformation (*DT*). In the era of the Internet, a great improvement in efficiency by using digital technology has attracted both traditional industries and emerging industries to obtain competitive advantages through digital transformation [68, 69]. The term 'digital transformation' is not simply a reference to the mining of an organization's data profile; rather, it denotes the digital empowerment of an organization's entire processes [70, 71]. Digital transformation is a comprehensive and methodical concept that delineates the utilization of novel digital technologies to attain substantial business growth, thereby optimizing customer experience, streamlining operational activities and synthesizing innovative business models [72]. In practice, digital transformation must address not only technological issues but also strategic issues, providing robust decision support for strategic issues [73]. However, the current academic approaches to measuring digital transformation are inadequate in terms of capturing the fundamental essence of this phenomenon. For instance, the questionnaire scoring approach [74, 75] and the text analysis approach [76] are insufficient in this regard. Accordingly, the digital transformation index adopted in this paper provides a comprehensive assessment of digital innovation across two perspectives: preparation and outcome and combines textual analysis and data analysis, specifically from six dimensions: strategic leadership, technology-driven, organizational empowerment, environmental support, digital achievements, and digital applications. The specific indicators and indicator weights are presented in the S1 Appendix. The index data source is the CSMAR database.

**3.2.3. Mediating variables.** Information Transparency (*IT*). Information transparency is gauged using accounting surplus transparency. Following Hutton, Marcus [77], a firm's information transparency is measured by the sum of the absolute values of manipulative accruals (Opaque) over the past three years; the higher the Opaque, the lower the transparency. The Opaque formula is as follows:

$$Opaque = \sum\nolimits_{i=1}^{3} |DisAcc_{t-i}| \tag{3}$$

Manipulative accruals (DisAcc) are estimated using the modified Jones model, represented by the following formula:

$$\frac{TA_{i,\,t}}{Asset_{i,\,t-1}} = a_1 \frac{1}{Asset_{i,\,t-1}} + a_2 \frac{\Delta REV_{i,\,t}}{Asset_{i,\,t-1}} + a_3 \frac{PPE_{i,\,t}}{Asset_{i,\,t-1}} + \varepsilon_{i,\,t} \tag{4}$$

$$DisAcc_{i,\,t} = \frac{TA_{i,\,t}}{Asset_{i,\,t-1}} - \left( \hat{a}_1 \frac{1}{Asset_{i,\,t-1}} + \hat{a}_2 \frac{\Delta REV_{i,\,t} - \Delta REC_{i,\,t}}{Asset_{i,\,t-1}} + \hat{a}_3 \frac{PPE_{i,\,t}}{Asset_{i,\,t-1}} \right) \tag{5}$$

Where TA is total accruals, Asset represents total assets, ΔREV is the growth in sales revenue, ΔREC is the growth in accounts receivable, and PPE is the original cost of fixed assets.

Internal Control (*IC*). The quality of internal control is measured using the "Internal Control Index of Listed Companies in China" published by the Shenzhen Dibao Database. Data are adjusted by subtracting 1000 during the regression analysis process.

**3.2.4. Threshold variables.** Technology accumulation (*INNO*). The number of annual enterprise patent applications, a common measure in academia for enterprise technological innovation [78], is used to gauge enterprise technology accumulation. Enterprise size (*TA*) is measured by total enterprise assets.

**3.2.5. Control variables.** To account for the impact of internal factors and external stakeholders on the internal informed trading behavior of the enterprise, the study includes six

**Table 1. Descriptive statistics.**

| Variable | Obs | Mean | Std. Dev. | Min | Max |
|---|---|---|---|---|---|
| VPIN | 20064 | 17.839 | 2.007 | 0 | 27.494 |
| DT | 20064 | 3.625 | 1.066 | 2.135 | 8.004 |
| IC | 20064 | 0.623 | 0.167 | 0 | 0.995 |
| IT | 20064 | 0.16 | 0.129 | 0 | 1.142 |
| INVO | 20064 | 45.071 | 331.867 | 0 | 14431 |
| TA | 20064 | 2.432e+10 | 1.078e+11 | 3083701 | 2.730e+12 |
| LR | 20064 | 2.14 | 3.119 | -5.132 | 190.869 |
| DAR | 20064 | 0.477 | 0.289 | -0.195 | 13.397 |
| ROE | 20064 | 0.06 | 6.065 | -186.557 | 713.204 |
| PIR | 20064 | 1.963 | 78.608 | -11.925 | 9290.911 |
| TobinQ | 20064 | 2.324 | 12.758 | 0.509 | 1056.725 |
| IIR | 20064 | 47.7 | 22.596 | 0 | 98.927 |

control variables: current ratio (*LR*), gearing ratio (*DAR*), *ROE*, growth rate of operating income (*PIR*), *TobinQ*, and institutional investor shareholding ratio (*IIR*).

## 3.3. Model construction

This study constructs a fixed effects model to examine the impact of enterprise digital transformation on the probability of informed transactions:

$$VPIN_{it} = \beta_0 + \beta_1 DT_{it} + \beta_2 Contorls_{it} + \mu_t + \gamma_s + \sigma_h + \varepsilon_{it} \tag{6}$$

where the dependent variable is the probability of informed transaction (*VPIN*), the independent variable is the digital transformation (*DT*) of enterprises, *Controls* is the matrix of control variables, $\varepsilon$ represents the model's random error term, $\mu_t$ denotes the time fixed effect, $\gamma_s$ represents the provincial fixed effect, and $\sigma_h$ signifies the industry fixed effect. In addition, all regressions control for clustered standard errors to mitigate bias and include controls for time, provincial, and industry fixed effects. Table 1 presents the descriptive statistics for each variable, and Table 2 displays the correlation matrix.

## 4. Empirical results and discussion

Based on Model 6, Table 3 presents the regression presents of firms' digital transformation on the probability of informed trading. Column (1) of Table 3 indicates that the regression coefficient of enterprise digital transformation on the probability of informed trading is significantly negative at the 1% level, with a coefficient of -0.415 and an $R^2$ of 0.356. Column (2) of Table 3

**Table 2. Correlation coefficient matrix.**

|  | DT | LR | DAR | ROE | PIR | TobinQ | IIR |
|---|---|---|---|---|---|---|---|
| DT | 1 | | | | | | |
| LR | 0.026*** | 1 | | | | | |
| DAR | -0.082*** | -0.356*** | 1 | | | | |
| ROE | -0.008 | 0.001 | -0.003 | 1 | | | |
| PIR | -0.012* | 0.017** | 0.002 | 0 | 1 | | |
| TobinQ | -0.011 | 0.025*** | 0.033*** | 0.052*** | 0.035*** | 1 | |
| IIR | -0.151*** | -0.106*** | 0.118*** | 0.001 | 0.003 | -0.021*** | 1 |

**Table 3. Baseline regression.**

|  | **(1)** | **(2)** |
|---|---|---|
|  | *VPIN* | *VPIN* |
| DT | -0.415*** | -0.410*** |
|  | (0.016) | (0.016) |
| LR |  | 0.029*** |
|  |  | (0.011) |
| DAR |  | -0.201* |
|  |  | (0.108) |
| ROE |  | -0.010*** |
|  |  | (0.004) |
| PIR |  | 0.000 |
|  |  | (0.000) |
| TobinQ |  | 0.004*** |
|  |  | (0.001) |
| IIR |  | -0.003*** |
|  |  | (0.001) |
| Year FE | Yes | Yes |
| Province FE | Yes | Yes |
| Industry FE | Yes | Yes |
| $R^2$ | 0.356 | 0.363 |
| N | 20064 | 20064 |

[a]Note: Robust standard errors in parentheses

*, ** and *** indicate significance levels of 10%, 5% and 1%, respectively (same as below).

reports the results after controlling for variables that could influence the dependent variable, revealing that the coefficient remains significantly negative at the 1% level, with a value of -0.410 and an $R^2$ of 0.363. This stabilization of significance and the notable improvement in model fit confirm the appropriateness of the selected control variables.

In markets, disparities in the accessibility of information and the elevated costs associated with market investors' access to information result in informed market trading [18, 38]. The digital transformation facilitates the immediacy, transparency, and verifiability of internal information transmission, which reduces the difficulty for insider to manipulate information [42, 79]. Concurrently, it guides the transformation of private to public information, reducing the cost of information collection for market participants while increasing the availability of information [43]. Furthermore, digital transformation encourages enterprises to enhance the uniformity of information delivery, which facilitates the release of more effective signals and bolsters the credibility of public information. And information transparency and efficient communication can reinforce internal management and reduce the occurrence of informed transactions [46]. All in all, the regression conclusion corroborates the theoretical analysis and substantiates Hypothesis 1.

Furthermore, the dataset is segmented into groups based on the geographical location, nature, and high-tech attributes of the firms to examine the impact heterogeneity of digital transformation on the probability of informed transactions. Columns (1) to (3) of Table 4 display the grouped regression results for firms in the Eastern, Central, and Western regions. The results indicate a significant inhibitory effect of digital transformation on the probability of informed trading across all regions. However, the inhibitory effect in western region is not as pronounced as in the Eastern and Central regions, suggesting a regional disparity in the

**Table 4. Analysis of regional heterogeneity.**

|  | (1) | (2) | (3) |
|---|---|---|---|
|  | *VPIN* | *VPIN* | *VPIN* |
| *DT* | -0.391*** | -0.472*** | -0.319*** |
|  | (0.019) | (0.040) | (0.048) |
| *District* | East | Middle | West |
| *Control Variable* | Yes | Yes | Yes |
| *Year FE* | Yes | Yes | Yes |
| *Province FE* | Yes | Yes | Yes |
| *Industry FE* | Yes | Yes | Yes |
| $R^2$ | 0.378 | 0.415 | 0.398 |
| *N* | 13282 | 4079 | 2702 |

effectiveness of digital transformation. This conclusion can be explained by the difference of digital infrastructure and human resources. Digital infrastructure encompasses a range of essential elements, including high-speed internet, cloud computing, and data analytics. These components are vital for the successful implementation of digital transformation. Compared with developing area such as western provinces, firms in developed area such as eastern and central provinces benefit from robust digital ecosystems that enhance governance, transparency, and market efficiency, thereby reducing information asymmetry and curbing informed trading [80]. With regard to human capital, disparities in regional skill levels are of considerable consequence. Compared with western province, the eastern and central provinces have a greater concentration of skilled labor in digital sectors, supported by more robust education and training systems, which enables firms to implement digital systems that reduce information asymmetry and prevent informed trading.

As indicated in columns (1) and (2) of Table 5, the coefficients of enterprise digital transformation on the probability of informed trading are significantly negative at the 1% level for both SOEs and non-SOEs, with values of -0.397 and -0.399, respectively, after accounting for time, provincial, and industry fixed effects. These results suggest that the inhibitory effect of digital transformation on informed trading behavior is more pronounced in non-SOEs than in SOEs. This conclusion can be explained from two distinct perspectives. Firstly, SOEs, which are driven by political objectives rather than profit maximization, face more significant agency issues. This is because the interests of managers may not align with those of owners or shareholders. This subsequently diminishes their incentive to comprehensively implement digital

**Table 5. Analysis of heterogeneity in nature and high-tech attributes of firms.**

|  | (1) | (2) | (3) | (4) |
|---|---|---|---|---|
|  | *VPIN* | *VPIN* | *VPIN* | *VPIN* |
| *DT* | -0.417*** | -0.379*** | -0.422*** | -0.387*** |
|  | (0.023) | (0.022) | (0.019) | (0.028) |
| *Character* | SOEs | Non-SOEs | High-tech | Non-high-tech |
| *Control Variable* | Yes | Yes | Yes | Yes |
| *Year FE* | Yes | Yes | Yes | Yes |
| *Province FE* | Yes | Yes | Yes | Yes |
| *Industry FE* | Yes | Yes | Yes | Yes |
| $R^2$ | 0.370 | 0.392 | 0.391 | 0.352 |
| *N* | 10268 | 9794 | 10859 | 9193 |

**Table 6.  Robust test (1).**

|  | (1) | (2) | (3) | (4) |
|---|---|---|---|---|
| DT | -0.414*** | -0.374*** | -0.206*** |  |
|  | (0.018) | (0.015) | (0.011) |  |
| SZ |  |  |  | -12.311*** |
|  |  |  |  | (2.727) |
| ontrol Variable | Yes | Yes | Yes | Yes |
| Year FE | Yes | Yes | Yes | Yes |
| Province FE | Yes | Yes | Stock FE | Yes |
| Industry FE | Yes | Yes | No | Yes |
| R2 | 0.322 | 0.387 | 0.080 | 0.343 |
| N | 20064 | 16720 | 20064 | 19778 |

transformation with the objective of enhancing governance and transparency. In contrast, non-SOEs, driven by profit and competition, have stronger incentives to leverage digital transformation to reduce information asymmetry and curb informed trading [81, 82]. Additionally, SOEs frequently operate within a culture that values stability and adherence to government objectives over innovation. This conservative approach may impede the adoption of digital transformation, which could in turn hinder its effectiveness in promoting transparency and preventing informed trading. In contrast, non-SOEs, particularly in competitive industries, foster a culture of innovation and efficiency, making them more likely to embrace digital transformation fully, including its governance and transparency benefits that help curb informed trading.

Columns (3) and (4) of Table 5 show that the coefficient of enterprise digital transformation on the probability of informed trading remains significantly negative at the 1% level for both high-tech and non-high-tech enterprises, with coefficients of -0.418 and -0.401, respectively, after controlling for the same fixed effects. Given the speculative nature of the high-tech industry and the advanced integration of digital technology within these firms, the impact on informed trading behavior is markedly stronger in high-tech enterprises compared to non-high-tech enterprises. This suggests that the mature application of digital technology in high-tech enterprises enhances their ability to mitigate informed trading behavior. High-tech firms are distinguished by their inherent technological expertise and advanced digital capabilities. These firms are often at the forefront of the adoption and development of sophisticated digital technologies, including big data analytics, artificial intelligence, and blockchain, which are instrumental in facilitating enhanced transparency and corporate governance [82]. The existing digital infrastructure facilitates the integration of new technologies that monitor, track, and regulate information flows within the firm, thereby reducing information asymmetry. Consequently, high-tech firms are better positioned to implement digital systems that restrict informed trading by ensuring the more widespread dissemination of information and its greater protection from exploitation for personal gain. In contrast, firms in other sectors may lack the requisite technological foundation for the effective implementation of digital transformation initiatives.

## 5. Robustness tests

The digital transformation of enterprises and their internal informed trading behavior are influenced by significant global and domestic events. For example, following a major adverse financial event, a firm's stock price volatility may affect informed trading behavior, and its digitalization process could be impeded. Analytical results that overlook such major events may

be prone to endogeneity, resulting in skewed conclusions. During the data sample timeframe of this study, two critical events occurred: the Chinese stock market crash in 2015 and the global COVID-19 pandemic in 2020. To address this, the regression analysis was re-conducted excluding data from 2015 and 2020. As shown in Column (1) of Table 6, the digital transformation of enterprises continues to inhibit the occurrence of informed trading behavior, aligning with the findings of the baseline model.

Extreme values resulting from anomalous changes in individual enterprises can skew the analysis of the impact of digital transformation on informed trading behavior. To mitigate this, the study winsorizes continuous variables by 99% to minimize the potential influence of these extremes. Column (2) of Table 6 shows that, after this adjustment, the coefficient of enterprise digital transformation on the probability of informed trading remains significantly negative at the 1% level, consistent with the baseline model findings.

The time frame of the sample used in this paper is from 2011 to 2022. However, the digital transformation of enterprises may have begun before 2011, which could result in a left truncation phenomenon in the data. To address this issue, this paper employs the Tobit model instead of the fixed-effects model for regression, as this approach is better suited to handle left truncation data. The results of the Tobit regression are presented Column (3) of Table 6 and we can find that the regression result is consist with the baseline model.

To further substantiate the robustness of the conclusions presented in the paper, this study employs a modified measurement approach to digital transformation. This approach utilizes the proportion of digital economy-related intangible assets relative to the total intangible assets of listed companies at the end of the year as a proxy variable. Specifically, in instances where an intangible asset line item contains keywords indicative of digital economy technology, such as "software," "network," "client," "management system," "intelligent platform," and so forth, as well as related patents, the line item is designated as "digital economy technology intangible assets". The results of the regression are presented Column (4) of Table 6 and we can find that the regression result is consist with the baseline model.

If the impact of dynamic panel data is not incorporated into the regression model, the potential lagged effect of digital transformation on informed transactions will be overlooked. Accordingly, this paper further considers the endogeneity issue that may be introduced by dynamic panels on the basis of the fixed-effect model, and employs the dynamic panel regression model to conduct causal testing. Column (1) of Table 7 presents the results of the dynamic panel regression, which demonstrate that the P-values of the AR test and the Sargan test are both greater than 0.1 indicating the absence of second-order autocorrelation and the coefficients of digital transformation is consist with the baseline model. These findings substantiate the robustness of the research presented in this paper.

In addition to the process of digitalization itself, the external environment also exerts an influence on the occurrence of informed transactions within a company. Accordingly, this paper incorporates two variables, namely financial supervision which is measured by ratio of financial regulatory expenditures to financial sector value added and media perspective which is measured by Janis-Fadner coefficient, into the regression model in order to control for the potential impact of the external environment. The results of the regression are presented Column (2) of Table 7 and we can find that the regression result is consist with the baseline model.

This research might still encounter endogeneity issues. To address this, the instrumental variable method is employed, using the industry average of the core explanatory variable, lagged by one period, as the instrumental variable. Columns (3) and (4) of Table 7 present the results of the first-stage and second-stage regressions, respectively. From column (3), the coefficients of the instrumental variables on digital transformation are significantly positive at the 1% level, the CD Wald F-statistic stands at 375.68, and the Stock-Wright LM S-statistic is

**Table 7. Robust test (2).**

| | (1) | (2) | (3) | (4) |
|---|---|---|---|---|
| DT | -2.055** | -0.396*** | | -0.530*** |
| | (0.825) | (0.016) | | (0.115) |
| L.VPIN | 0.499*** | | | |
| | (0.166) | | | |
| FR | | -1.864 | | |
| | | (1.699) | | |
| Media | | -0.097*** | | |
| | | (0.033) | | |
| IV | | | 0.668*** | |
| | | | (0.034) | |
| ontrol Variable | Yes | Yes | Yes | Yes |
| Year FE | Yes | Yes | Yes | Yes |
| Province FE | Stock FE | Yes | Yes | Yes |
| Industry FE | No | Yes | Yes | Yes |
| R2 | 0.387 | 0.367 | | 0.042 |
| N | 18392 | 19667 | 18392 | 18392 |
| AR(2) | 0.809 | | | |
| Sargan | 0.998 | | | |
| D Wald F | | | 375.68 | |
| SW S stat. | | | 20.50*** | |

significant at the 1% level, indicating that the chosen instrumental variable does not suffer from weakness. Column (4) reveals that the regression coefficient of digital transformation on the probability of informed trading is significantly negative, mirroring the benchmark regression results and underscoring the robustness of this study's regressions.

## 6. Further research and discussion

### 6.1. Impact mechanisms

Tables 8 and 9 explore the results of the chained mediation estimation regarding the impact of firms' digital transformation on informed trading. Column (1) of Table 8 shows that the

**Table 8. Mediating effects test.**

| | (1) | (2) | (3) |
|---|---|---|---|
| | IT | IC | VPIN |
| DT | -0.009*** | | |
| | (0.001) | | |
| IT | | -0.164*** | |
| | | (0.012) | |
| IC | | | -2.264*** |
| | | | (0.100) |
| Control Variable | Yes | Yes | Yes |
| Year FE | Yes | Yes | Yes |
| Province FE | Yes | Yes | Yes |
| Industry FE | Yes | Yes | Yes |
| $R^2$ | 0.118 | 0.153 | 0.372 |
| N | 20064 | 20064 | 20064 |

**Table 9. Bootstrap test results for chained mediation effects.**

|  | Effect | BootSE | BootLLCI | BootULCI |
|---|---|---|---|---|
| Direct Effect |  |  |  |  |
| DT→VPIN | -0.324 | 0.013 | -0.350 | -0.298 |
| Indirect Effect |  |  |  |  |
| Total | -0.018 | 0.002 | -0.023 | -0.015 |
| DT→IT→VPIN | -0.002 | 0.001 | -0.003 | -0.001 |
| DT→IC→VPIN | -0.016 | 0.002 | -0.019 | -0.012 |
| DT→IT→IC→VPIN | -0.001 | 0.000 | -0.002 | -0.001 |

[a]Note: Total (*) indicates total indirect effects.

coefficient of enterprise digital transformation on enterprise information transparency is significantly negative at the 1% level, indicating that digital transformation significantly enhances information transparency. Column (2) reveals that the coefficient of enterprise information transparency on the internal control index is significantly negative at the 1% level, suggesting a notable internal control enhancement effect. Column (3) demonstrates that the coefficient of enterprise internal control on the probability of informed trading is significantly negative at the 1% level, indicating that enhanced internal control significantly reduces the occurrence of informed trading behavior.

The chain mediation test on the impact of enterprise digital transformation on informed trading employs the Process plug-in in SPSS software, using the Bootstrap method with samples repeated 5,000 times and a 95% confidence interval. The results, shown in Table 9, reveal that the coefficient of the direct effect is -0.324, with a confidence interval of [-0.350, -0.298], confirming its significance. The path "enterprise digital transformation → information transparency→ probability of informed transactions" has a coefficient of -0.002, with a confidence interval of [-0.003, -0.001], confirming its significance. The path "enterprise digital transformation → internal control → probability of informed transactions" has a coefficient of -0.016, with a confidence interval of [-0.019, -0.012], confirming its significance. The complete mediation path "enterprise digital transformation → information transparency→ internal control → probability of informed transactions" has a coefficient of -0.001, with a confidence interval of [-0.002, -0.001], confirming its significance. Additionally, the total chain mediation effect has a coefficient of -0.018, with a confidence interval of [-0.023, -0.015], confirming that information transparency and internal control are significant chain mediating variables.

As demonstrated by the preceding theoretical analysis, the confluence of information asymmetry and agency issues facilitates the circumvention of management regulations by directors, supervisors, and senior executives. This, in turn, gives rise to internal control deficiencies and heightens the probability of management's informed transactions [47]. The digital transformation of enterprises can enhance transparency of enterprise information, thereby strengthening internal control capabilities and inhibiting the possibility of informed transactions. On the one hand, digital transformation can optimize information channels, increase the supply of enterprise decision-making information, improve decision-making efficiency, enhance the degree of information disclosure, and mitigate information asymmetry [48]. Conversely, digital transformation not only facilitates shareholders' participation in management decision-making, improves the effectiveness of information communication, and reduces agency problems between shareholders and managers, but also promotes the optimization of organizational structure, improves management efficiency, avoids internal control norms, and inhibits managers' informed transactions [51, 53, 55]. Therefore, the regression results are consistent with

the theory, suggesting that digital transformation enhances enterprise information transparency, which in turn strengthens internal control capabilities, thereby reducing the probability of informed trading behavior, thus establishing Hypothesis 2.

## 6.2. Threshold effect

To examine whether there is a threshold effect in the impact of digital transformation on informed trading behavior, this paper employs a dynamic panel threshold model using technology accumulation and enterprise size as threshold variables, following Hansen [83] for model construction. The model is structured as follows:

$$VPIN_{it} = \beta_0 + \beta_1 VPIN_{i,t-1} + \beta_{21} DT_{it} F(X_{it} \leq \gamma_1) + \beta_{22} DT_{it} F(X_{it} > \gamma_2) + \beta_3 Controls_{it} + \varphi_i + \varepsilon_{it} \ (7)$$

Specifically, γ represents the threshold value, $X_{it}$ is the threshold variables are technology accumulation or total assets. F is a binary function that equals 1 when the specified conditions are met and 0 otherwise. Technology accumulation is quantified by the annual number of patent applications filed by the enterprise, and total assets are measured by the enterprise's total assets.

Based on Model 7, this study investigates the dynamic panel threshold effect of digital transformation on informed trading behavior and results displayed in Table 10. When technology accumulation is the threshold variable and is less than or equal to 56, the regression coefficient is -0.295 but not significant, indicating that digital transformation cannot significantly inhibits informed trading behaviors when has less innovation accumulation. When technology accumulation exceeds 56, the coefficient is -1.016 and significant at the 1% level, suggesting a stronger inhibitory effect of digital transformation on informed trading behaviors. With total assets

**Table 10. Threshold regression results.**

| | | (1) | | (2) |
|---|---|---|---|---|
| | | Vpin | | Vpin |
| LR | | 2.091*** | | 1.721*** |
| | | (0.583) | | (0.492) |
| DAR | | 6.756 | | 5.812 |
| | | (8.222) | | (8.935) |
| ROE | | -0.448 | | -0.614** |
| | | (0.315) | | (0.313) |
| PIR | | 0.0335 | | 0.0228 |
| | | (0.0408) | | (0.0365) |
| TobinQ | | -1.122*** | | -1.210*** |
| | | (0.235) | | (0.218) |
| IIR | | 0.0122 | | 0.0166 |
| | | (0.0403) | | (0.0323) |
| DT | INNO≤56 | -0.295 | TA≤3.690e+09 | 0.160 |
| | | (0.348) | | (0.509) |
| DT | INNO>56 | -1.016*** | TA >3.690e+09 | -0.476* |
| | | (0.365) | | (0.262) |
| L.Vpin | | 0.117** | | 0.0978 |
| | | (0.0579) | | (0.0855) |
| N | | 18392 | | 18392 |
| AR(1) | | 0.162 | | 0.200 |
| AR(2) | | 0.200 | | 0.199 |

as the threshold variable, when total assets are less than or equal to 3.690e+09, the coefficient is 0.160 but not significant, indicating that digital transformation cannot significantly inhibits informed trading behaviors when has less asset. When it exceeds 3.690e+09, the coefficient is -0.476 and significant at the 10% level, showing significant suppression of informed trading behavior on informed trading behaviors. The regression results demonstrate that there is a notable threshold effect on the inhibition of informed transactions by digital transformation. The inhibition effect of informed transactions can be observed only when the technology accumulation or asset scale of enterprises reaches a specific threshold, at which point they possess considerable technical experience and sufficient technical personnel, as well as a robust internal organization capable of providing substantial support in the digital transformation of enterprises. Therefore, hypotheses 3a and 3b are supported by the result.

## 7. Research conclusions, policy recommendations and limitations

Due to information asymmetry and market imperfections, informed trading varies across markets, particularly in emerging market countries where market systems are underdeveloped, increasing the likelihood of such trading. Informed trading can mislead market prices, disrupt resource allocation, and trigger adverse selection transactions that heighten market liquidity risks, potentially precipitating market downturns and impeding healthy market development. Therefore, in the rapidly evolving digital economy, examining the impact of enterprise digital transformation on informed trading behavior is vitally important. This study uses a fixed-effects model to analyze the influence of corporate digital transformation on the probability of informed trading, employing data from all A-share listed companies from 2011 to 2022. The findings reveal that digital transformation substantially reduces the likelihood of informed trading, a result consistent with the baseline model and substantiated by a series of robustness tests, thereby affirming the reliability of this research. Further exploration of heterogeneity shows that compared to state-owned and non-high-tech enterprises and enterprises in the western region, digital transformation has a more pronounced inhibitory effect on informed trading behavior in non-state-owned and high-tech enterprises and enterprises in the eastern and central regions. Through theoretical discussion and model validation of the chain mediation effect, it is established that "digital transformation enhances information transparency, mitigates internal control deficiencies, strengthens internal control capabilities, and thereby reduces the probability of informed trading." Furthermore, this study examines the dynamic panel threshold mechanism and determines that digital transformation can only significantly inhibit the informed transactions when enterprises have reached a certain level of technological and asset accumulation.

Combining the above arguments and test results, the following recommendations are made: First, enhance support for the digital transformation of non-state-owned and high-tech enterprises by introducing relevant policies and measures such as tax incentives, financial subsidies, and special funds. This will incentivize them to accelerate their digital transformation. Simultaneously, strengthen policy publicity and training to increase awareness of the importance of digital transformation within these enterprises, guiding them to effectively utilize digital tools to improve management efficiency and information transparency. Second, promote the balanced development of inter-regional digital infrastructure. Given the disparities in digital technology availability across regions, the government should increase investment in digital infrastructure in less-developed areas, such as broadband networks, data centers, and cloud computing platforms. This will help narrow the digital divide and encourage cross-regional technological exchanges and cooperation, facilitating the transfer of technological achievements from advanced to less-developed regions. A balanced development of digital

infrastructure lays the groundwork for enterprise digital transformation, creating a conducive environment for digital transformation across all regions, thus enhancing the information transparency of the overall market and reducing the occurrence of informed trading. Third, strengthen enterprise internal control and the information disclosure system. Improve relevant laws and regulations, enhance the construction of enterprise internal control mechanisms, and require enterprises to establish a robust information disclosure system to ensure the truthfulness, accuracy, and timeliness of information. Concurrently, intensify the supervision and inspection of enterprise internal control implementations, and curb violations of information disclosure regulations and internal control requirements. Finally, implement differentiated digital transformation strategies for enterprises of varying sizes and development stages. When an enterprise has limited technological accumulation and a smaller scale, it should focus on its own growth and technological innovation to lay a solid foundation for leveraging the advantages of digital technology development.

There are also some limitations in this study. Firstly, it should be noted that the research sample presented in this paper is based on the market environment in China. However, it is important to acknowledge that there are differences in the process and mode of digitalization across different countries worldwide. Consequently, it would be beneficial to conduct further in-depth research from a global perspective in the future, which would enhance the universality of the research. Secondly, although the mediation effect and threshold effect are discussed, there are still some external environmental factors that may have an impact on the main effect, which can be further discussed. Thirdly, VPIN is used for the measurement of informed transaction behavior in this paper, which can be deepened in the future, so as to further study the impact of digital transformation on informed transaction behavior with different motivations. Lastly, the study covers data from 2011 to 2022, but the onset of digital transformation may have varied across firms, with some adopting digital tools much earlier or later than others. The study's cross-sectional analysis might not fully capture the dynamic and evolving nature of digital transformation, which could influence the results. A more detailed longitudinal analysis could provide deeper insights into the temporal effects of digitalization on informed trading.

## Supporting information

**S1 Appendix. Digital transformation indicator system.**
(DOCX)

## Author Contributions

**Conceptualization:** Hualing Wang.

**Data curation:** Hualing Wang.

**Formal analysis:** Hualing Wang.

**Funding acquisition:** Hualing Wang.

**Investigation:** Hualing Wang.

**Methodology:** Hualing Wang.

**Writing – original draft:** Hualing Wang.

**Writing – review & editing:** Hualing Wang.

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
