## [Decision Letter · Decision Letter 0]

27 Sep 2024

PONE-D-24-38638A Study on the Impact of Enterprise Digital Transformation on Informed TradingPLOS ONE

Dear Dr. Wang,

Thank you for submitting your manuscript to PLOS ONE. After careful consideration, we feel that it has merit but does not fully meet PLOS ONE’s publication criteria as it currently stands. Therefore, we invite you to submit a revised version of the manuscript that addresses the points raised during the review process. An interesting paper on a relevant topic. Two qualified reviewers in this area of research have commented on your above paper and identified various shortcomings that need to be addressed. They indicated that it is not acceptable for publication in its present form, but it may be acceptable after some significant revision. Upon my review of the manuscript, I concur with the reviewers' comments. I ask the author(s) to review each comment below and submit a revised version of the manuscript.

We look forward to receiving your revised manuscript.

Kind regards,

Yaman Roumani

Academic Editor

PLOS ONE

Journal Requirements:

“This research is supported by Fujian Social Science Fund Project (FJ2024B035)”

Comments from PLOS Editorial Office: We note that one or more reviewers has recommended that you cite specific previously published works. As always, we recommend that you please review and evaluate the requested works to determine whether they are relevant and should be cited. It is not a requirement to cite these works. We appreciate your attention to this request.

Reviewers' comments:

Reviewer's Responses to Questions

**Comments to the Author**

1. Is the manuscript technically sound, and do the data support the conclusions?

Reviewer #1: Partly

Reviewer #2: Yes

2. Has the statistical analysis been performed appropriately and rigorously? 

Reviewer #1: No

Reviewer #2: Yes

3. Have the authors made all data underlying the findings in their manuscript fully available?

Reviewer #1: Yes

Reviewer #2: Yes

4. Is the manuscript presented in an intelligible fashion and written in standard English?

Reviewer #1: No

Reviewer #2: Yes

5. Review Comments to the Author

Reviewer #1: Over-reliance on Descriptive Findings: The paper heavily focuses on descriptive results of how digital transformation impacts informed trading, but the analysis lacks a deeper exploration of the mechanisms driving these outcomes. While the mediating variables are discussed, a more nuanced examination of why and how these factors interact would strengthen the findings.

Lack of Granularity in Digital Transformation Definition: The paper does not clearly define what constitutes "digital transformation" in terms of specific metrics or indices. Although digital transformation is a key term, there is limited discussion on how it is measured, particularly in different industry sectors or regions.

Generalization Across Different Regions and Enterprise Types: The manuscript discusses the effects of digital transformation in various regions and enterprise types (e.g., state-owned vs. non-state-owned enterprises), but it lacks a detailed analysis of the potential reasons for regional disparities. More depth is required to explain why some regions or enterprise types experience more pronounced effects than others.

Limited Use of Robustness Checks: While the manuscript includes some robustness tests, additional checks could be implemented to further validate the results, such as alternative models or different assumptions regarding the nature of the data. This would provide more confidence in the reliability of the conclusions.

Absence of Alternative Explanations: The paper does not explore alternative explanations for the reduction in informed trading beyond digital transformation. Other macroeconomic or regulatory factors could also influence these results, and their exclusion may limit the completeness of the analysis.

Methodological Gaps: The use of the fixed-effect model, while appropriate for this type of analysis, could be supplemented with other methodologies, such as dynamic panel models, to capture potential lag effects of digital transformation on informed trading over time.

Lack of Discussion on Limitations: The manuscript lacks a dedicated section that clearly outlines the limitations of the study. Issues such as data availability, potential biases in measurement, or regional differences in digital infrastructure are briefly mentioned but not discussed in sufficient detail.

Reviewer #2: 1. In the 4th paragraph in the introduction part, the following sentence sould be added the information in the quatation mark with the suggested citation because nowadays social media and wedsite is becominig interesting issue.

Moreover, digital technology can also suppress the supply of noise

information and enhance the quality of public information, "especially representational information on the website and social media."

suggested citation:

Hiranphaet, A., Sooksai, T., Aunyawong, W., Poolsawad, K., Shaharudin, M.R., & Siliboon, R. (2022). Development of value chain by creating social media for disseminating marketing content to empower potential of participatory community-based tourism enterprises. International Journal of Mechanical Engineering, 7(5), 431-437.

2. Revise both in-text citation and references as suggested below by the journal:

References are listed at the end of the manuscript and numbered in the order that they appear in the text. In the text, cite the reference number in square brackets (e.g., “We used the techniques developed by our colleagues [19] to analyze the data”). PLOS uses the numbered citation (citation-sequence) method and first six authors, et al.

You can see the following link: https://journals.plos.org/plosone/s/submission-guidelines#loc-references

3. The discussion is not founded in the article so please see the following guidlines from the journal:

Results, Discussion, Conclusions

These sections may all be separate, or may be combined to create a mixed Results/Discussion section (commonly labeled “Results and Discussion”) or a mixed Discussion/Conclusions section (commonly labeled “Discussion”). These sections may be further divided into subsections, each with a concise subheading, as appropriate. These sections have no word limit, but the language should be clear and concise.

Together, these sections should describe the results of the experiments, the interpretation of these results, and the conclusions that can be drawn.

Authors should explain how the results relate to the hypothesis presented as the basis of the study and provide a succinct explanation of the implications of the findings, particularly in relation to previous related studies and potential future directions for research.

6. PLOS authors have the option to publish the peer review history of their article (what does this mean?). If published, this will include your full peer review and any attached files.

Reviewer #1: **Yes: **Anak Agung Gde Satia Utama

Reviewer #2: No

---

## [Author Response · Author response to Decision Letter 0]

22 Oct 2024

(respond letter are upload as a word file)

Point-by-point responses to reviewer’ comments

Reviewer Comments:

Reviewer #1: 

Over-reliance on Descriptive Findings: The paper heavily focuses on descriptive results of how digital transformation impacts informed trading, but the analysis lacks a deeper exploration of the mechanisms driving these outcomes. While the mediating variables are discussed, a more nuanced examination of why and how these factors interact would strengthen the findings.

Reply: Thank you for your comments on the in-depth discussion of empirical analysis. This paper has a problem in the empirical analysis part as you said, that is, it relies on descriptive analysis but lacks theoretical analysis. Therefore, this study focuses on the combination of empirical results and theoretical analysis (e.g., information asymmetry theory and principal-agent theory), and the presentation of empirical results with appropriate theoretical analysis, so as to achieve more theoretical depth of the research results. The main effect analysis, mediation effect analysis and threshold effect analysis of this study have been modified and improved.

Lack of Granularity in Digital Transformation Definition: The paper does not clearly define what constitutes "digital transformation" in terms of specific metrics or indices. Although digital transformation is a key term, there is limited discussion on how it is measured, particularly in different industry sectors or regions.

Reply: Thank you for your comments on the digital transformation indicators! Digital transformation is the key indicator of this paper's research, and your comments are very important to improve the rigor of this paper's research. Based on your comments, I have added a definition of digital transformation, and also discussed the more commonly used metrics based on a large number of related studies, leading to the Digital Transformation Index (DTI) used in this paper based on a critical discussion (you can check the original page P16 for details). Since the research sample of this paper covers several industries and the vast majority of provinces and regions in China, the measurement of digital transformation is considered in terms of common characteristics across industries and regions. The index system and related weights of the Digital Transformation Index are shown in the Appendix (on page P37 of the original article).

Generalization Across Different Regions and Enterprise Types: The manuscript discusses the effects of digital transformation in various regions and enterprise types (e.g., state-owned vs. non-state-owned enterprises), but it lacks a detailed analysis of the potential reasons for regional disparities. More depth is required to explain why some regions or enterprise types experience more pronounced effects than others.

Reply: Thank you for your comments on deepening the analysis of heterogeneity! This paper also has the problem of relying on descriptive analysis, as you said and mentioned in the part of heterogeneity analysis, and does not analyze the essential causes of heterogeneity. According to your comments, this paper explains geographic location heterogeneity from the perspective of digital infrastructure and human resources, firm nature heterogeneity from the perspective of agency and organizational differences, and high-tech firm heterogeneity from the perspective of digital maturity, forming a deeper empirical analysis.

Limited Use of Robustness Checks: While the manuscript includes some robustness tests, additional checks could be implemented to further validate the results, such as alternative models or different assumptions regarding the nature of the data. This would provide more confidence in the reliability of the conclusions.

Reply: Thanks for your comments on the robustness tests! Based on your comments, this paper adds to the robustness test. First, the data years of this paper are from 2011 to 2022, but the digital transformation can be carried out before 2011, so the data may have left truncation characteristics, so this paper changes the fixed effect model to Tobit model for regression. Second, from the indicator measurement, this paper adopts the ratio of digital economy related intangible assets to the total intangible assets of listed companies at the end of the year as a proxy variable, and then conducts regression analysis. The robustness test results of the two methods are consistent with the baseline model, which further validates the robustness of the study in this paper. (Column 3 and 4 of Table 6 show the change in model and the change in variable measurement, respectively).

Table 6 Robust test (1)

 (1) (2) (3) (4)

DT -0.414*** -0.374*** -0.206*** 

 (0.018) (0.015) (0.011) 

SZ -12.311***

 (2.727)

Control Variable Yes Yes Yes Yes

Year FE Yes Yes Yes Yes

Province FE Yes Yes Stock FE Yes

Industry FE Yes Yes No Yes

R2 0.322 0.387 0.080 0.343

N 20064 16720 20064 19778

Absence of Alternative Explanations: The paper does not explore alternative explanations for the reduction in informed trading beyond digital transformation. Other macroeconomic or regulatory factors could also influence these results, and their exclusion may limit the completeness of the analysis.

Reply: Thank you for your comments on the impact of macroeconomic and external factors. Informed trading behavior within a firm receives the influence of the external environment, and if the analysis is conducted without controlling the influence of the external environment, it will have an impact on the robustness of the analysis results. Therefore, combining the opinions of the reviewing experts, this paper adds the control of external influence in the robustness type test, and includes two external factors, namely the intensity of financial regulation and the intensity of media supervision, in the regression model for regression, and the regression results can be seen to be consistent with the baseline model, which verifies the robustness of the research in this paper. (Column 2 of Table 7 shows the robustness test for the control of external influences)

Table 7 Robust test (2)

 (1) (2) (3) (4)

DT -2.055** -0.396*** -0.530***

 (0.825) (0.016) (0.115)

L.VPIN 0.499*** 

 (0.166) 

FR -1.864 

 (1.699) 

Media -0.097*** 

 (0.033) 

IV 0.668*** 

 (0.034) 

Control Variable Yes Yes Yes Yes

Year FE Yes Yes Yes Yes

Province FE Stock FE Yes Yes Yes

Industry FE No Yes Yes Yes

R2 0.387 0.367 0.042

N 18392 19667 18392 18392

AR(2) 0.809 

Sargan 0.998 

CD Wald F 375.68 

SW S stat. 20.50*** 

Methodological Gaps: The use of the fixed-effect model, while appropriate for this type of analysis, could be supplemented with other methodologies, such as dynamic panel models, to capture potential lag effects of digital transformation on informed trading over time.

Reply: Thank you for your suggestion about dynamic panels. In the research question of this paper, digital transformation and informed trading behavior in previous years will have a potential impact on informed trading behavior in the current year, which can only be further investigated by using dynamic panel model. Therefore, first, this study complements the robustness test using the dynamic panel model, from which the results of the dynamic panel model show that informed trading behavior in previous years will have a positive impact on informed trading behavior in the current year, except for digital transformation, which inhibits informed trading behavior in line with the baseline model. Second, the model is optimized as a dynamic panel threshold effect model in the threshold effect part, considering the impact of threshold effect under the dynamic panel factors, and it is known from the dynamic threshold model that digital transformation can play a role in inhibiting informed trading behavior only when the technological accumulation and asset accumulation of the enterprise reach a certain level. (Column 1 of Table 7 shows the robustness test for the dynamic panel, and Table 10 shows the regression results based on the dynamic panel threshold model).

Table 10 Threshold Regression Results

 (1) (2)

 Vpin Vpin

LR 2.091*** 1.721***

 (0.583) (0.492)

DAR 6.756 5.812

 (8.222) (8.935)

ROE -0.448 -0.614**

 (0.315) (0.313)

PIR 0.0335 0.0228

 (0.0408) (0.0365)

TobinQ -1.122*** -1.210***

 (0.235) (0.218)

IIR 0.0122 0.0166

 (0.0403) (0.0323)

DT INNO≤56 -0.295 TA≤3.690e+09 0.160

 (0.348) (0.509)

DT INNO>56 -1.016*** TA >3.690e+09 -0.476*

 (0.365) (0.262)

L.Vpin 0.117** 0.0978

 (0.0579) (0.0855)

N 18392 18392

AR(1) 0.162 0.200

AR(2) 0.200 0.199

Lack of Discussion on Limitations: The manuscript lacks a dedicated section that clearly outlines the limitations of the study. Issues such as data availability, potential biases in measurement, or regional differences in digital infrastructure are briefly mentioned but not discussed in sufficient detail.

Reply: Thank you for your comments regarding the discussion of limitations. The article's discussion of limitations is an important part of the article and a basis for more in-depth research in the future. This article explores the limitations of this article from three perspectives: generalizability, discussion of the external environment, and measurement of variables.

Reviewer #2: 

1. In the 4th paragraph in the introduction part, the following sentence sould be added the information in the quatation mark with the suggested citation because nowadays social media and wedsite is becominig interesting issue. Moreover, digital technology can also suppress the supply of noise information and enhance the quality of public information, "especially representational information on the website and social media."

suggested citation:

Hiranphaet, A., Sooksai, T., Aunyawong, W., Poolsawad, K., Shaharudin, M.R., & Siliboon, R. (2022). Development of value chain by creating social media for disseminating marketing content to empower potential of participatory community-based tourism enterprises. International Journal of Mechanical Engineering, 7(5), 431-437.

Reply: Thank you for your comments. Social media and websites play an important role in the dissemination of information, and adding a statement about social media and websites to this paragraph could better relate to the reality of the issue.

2. Revise both in-text citation and references as suggested below by the journal:

References are listed at the end of the manuscript and numbered in the order that they appear in the text. In the text, cite the reference number in square brackets (e.g., “We used the techniques developed by our colleagues [19] to analyze the data”). PLOS uses the numbered citation (citation-sequence) method and first six authors, et al.

You can see the following link: https://journals.plos.org/plosone/s/submission-guidelines#loc-references

Reply: Thank you for your comments! The citation of the article has been changed according to the relevant information you gave us and it meets the formatting requirements of PLOS journals.

3. The discussion is not founded in the article so please see the following guidlines from the journal:

Results, Discussion, Conclusions

These sections may all be separate, or may be combined to create a mixed Results/Discussion section (commonly labeled “Results and Discussion”) or a mixed Discussion/Conclusions section (commonly labeled “Discussion”). These sections may be further divided into subsections, each with a concise subheading, as appropriate. These sections have no word limit, but the language should be clear and concise.

Together, these sections should describe the results of the experiments, the interpretation of these results, and the conclusions that can be drawn.

Authors should explain how the results relate to the hypothesis presented as the basis of the study and provide a succinct explanation of the implications of the findings, particularly in relation to previous related studies and potential future directions for research.

Reply: Thank you for your comments! In the first draft of this study, the data were only reported based on the results of regression analysis, focusing on the descriptive analysis, but not sufficiently discussing and interpreting the regression results in relation to the theory. According to your suggestion, the title “Empirical Results” of this paper will be changed to "Empirical Results and Discussion", and the descriptive and theoretical analyses (principal-agent theory and information asymmetry theory) will be combined into main, mediation, and threshold effects, so that the results can be better discussed and interpreted.

---

## [Editor Report · Decision Letter 1]

29 Oct 2024

A Study on the Impact of Enterprise Digital Transformation on Informed Trading

PONE-D-24-38638R1

Dear Dr. Wang,

We’re pleased to inform you that your manuscript has been judged scientifically suitable for publication and will be formally accepted for publication once it meets all outstanding technical requirements.

Kind regards,

Yaman Roumani

Academic Editor

PLOS ONE
---

## [Editor Report · Acceptance letter]

13 Dec 2024

PONE-D-24-38638R1 

PLOS ONE

Dear Dr. Wang, 

I'm pleased to inform you that your manuscript has been deemed suitable for publication in PLOS ONE. Congratulations! Your manuscript is now being handed over to our production team.

Kind regards, 

on behalf of

Dr. Yaman Roumani 

Academic Editor

PLOS ONE